# Computational Study of the Propeller Position Effects in Wing-Mounted, Distributed Electric Propulsion with Boundary Layer Ingestion in a 25 kg Remotely Piloted Aircraft

José Ramón Serrano *, Andrés Omar Tiseira, Luis Miguel García-Cuevas and Pau Varela

CMT—Motores Térmicos, Universitat Politècnica de València, 46022 Valencia, Spain; anti1@mot.upv.es (A.O.T.); luiga12@mot.upv.es (L.M.G.-C.); pavamar@mot.upv.es (P.V.)
* Correspondence: jrserran@mot.upv.es; Tel.: +34-963877650

**Abstract:** Distributed electric propulsion and boundary layer ingestion are two attractive technologies to reduce the power consumption of fixed wing aircraft. Through careful distribution of the propulsive system elements, higher aerodynamic and propulsive efficiency can be achieved, as well as a lower risk of total loss of aircraft due to foreign object damage. When used on the wing, further reductions of the bending moment on the wing root can even lead to reductions of its structural weight, thus mitigating the expected increase of operating empty weight due to the extra components needed. While coupling these technologies in fixed-wing aircraft is being actively studied in the big aircraft segment, it is also an interesting approach for increasing the efficiency even for aircraft with maximum take-off masses as low as 25 kg, such as the A3 open subcategory for civil drones from EASA. This paper studies the effect of changing the propellers' position in the aerodynamic performance parameters of a distributed electric propulsion with boundary layer ingestion system in a 25 kg fixed-wing aircraft, as well as in the performance of the propellers. The computational results show the trade-offs between the aerodynamic efficiency and the propeller efficiency when the vertical position is varied.

**Keywords:** distributed electric propulsion; boundary layer ingestion; propeller; fixed wing

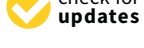



## 1. Introduction

Nowadays, some of the most important challenges related to the development and operation of new small aircraft, whether they are remotely piloted (RPAS) or autonomous, are their safety and environmental impact. Indeed, this is stated by different aviation safety agencies, such as in in the "Study on the societal acceptance of Urban Air Mobility in Europe" by the European Union Aviation Safety Agency (EASA) [1]. This information combined with the available data that predicts a broad growth of this type of aircraft in the coming years, reaching hundreds of thousands of units according to the Single European Sky ATM Research (SESAR) [2], are clear indicators of the need to seek solutions in the design of new aircraft, focused on reducing fuel and energy use and pollutant emissions. One way to achieve the efficiency objectives is by optimising the power train and aerodynamic design, researching more efficient aircraft models, according to NASA in the Information Technology Development Solutions (ITDS) of the Environmentally Responsible Aviation (ERA) project [3].

One of the most studied technologies in recent years to reduce pollutant emissions while maximizing the operational range and endurance of the aircraft is electric hybridisation (HE). It is worth mentioning the research of Auesser et al. [4], where they integrate and validate a parallel HE propulsion system for RPAS; the work of Harmon et al. [5], who proposed an optimisation of both aerodynamics and the HE propulsive system; or Kim et al. [6], who focused in the fuel economy optimisation of parallel HE. Through

hybridisation, decoupling between ICE and propeller shaft can be achieved, allowing integration of novel configurations such as distributed electric propulsion (DEP) and boundary layer ingestion (BLI).

DEP consists of separating the total electric power plant and propellers into smaller ones distributed along the wingspan. This distribution presents multiple advantages as increased flow circulation on the wing, increasing this way the aerodynamic efficiency; the use of the power plant as a control platform, reducing the need for aerodynamic actuators; easier maintenance of electric motors and their possible replacement; engine failure is less critical and the noise footprint can be reduced. The advantages of using DEP have been studied in the past years both for conventional and small aircraft [7,8].

The optimisation of a DEP system depends on the location of the electric engines, since its location and distribution have shown to have an important effect on the aircraft aerodynamics. However, there is a position of special interest: By distributing the propellers near the wing trailing edge it is possible to take advantage of the ingestion of the boundary layer produced by the wing. This phenomenon is known as boundary layer ingestion (BLI) and is widely studied in [9–11]. BLI is based on flux re-acceleration around the wing due to the ingestion produced by the propeller. This re-acceleration leads to a wake reduction, with a part of the propeller facing a lower incident air speed compared to what it would have in a classic configuration on the leading edge. In such a way, the engine needs less power for a given thrust as described by Budziszewski in [12], increasing the propulsive efficiency. At the same time, the ingestion could modify the airflow circulation around the airfoil, incrementing the intensity of the suction peak near the leading edge and reducing lift-induced drag, yet increasing the skin friction drag due to the acceleration, which translates into changes in the aerodynamic efficiency. In some studies, such as the work presented by Hall et al. [13], it is pointed out that the aerodynamic efficiency will be higher in the BLI configuration; likewise, Martínez et al. and Teperin et al. [14–16] come to the same conclusion in their BLI research applied to fuselages. By coupling the BLI configuration with DEP, a greater part of the wing and the propulsive plant is affected by the benefits of boundary layer ingestion. This allows, as shown in the work of Goldberg et al. [17], to decrease the fuel consumption compared with a classic distribution.

The main contribution of the current research paper is to analyse the possible positive effects on both aerodynamic and propulsive efficiency thanks to the use of distributed electric propulsion and boundary layer ingestion simultaneously, with a special emphasis on the analysis of the relative position between the propeller and the airfoil trailing edge, in the case of a small fixed-wing aircraft. The research is performed employing computational tools, with experimental validation whenever is possible.

The document is organized as follows. First, in Section 2 the main aircraft configuration is selected. Then, in Section 3, the main methods and models are presented, explaining in detail the modeling of the actuator disc. In Section 4 the different results are computed for different propeller positions, and the best case configuration is compared with a classical configuration without DEP or BLI. Finally, all main results and discussions are summarized in the conclusions in Section 5.

## 2. Design and Component Selection

In this section, the different design parameters and components are selected in order to prepare the simulations.

One of the main characteristics that limit the design of the aircraft, both aerodynamically and propulsively, is its size. In this case, the size of the aircraft is chosen to take into account the Spanish regulation for RPAS civil use [18] that fixes the maximum takeoff mass (MTOM) of civil RPAS (without special permits) in 25 kg at most, similarly to other RPAS regulations across Europe. Looking for commercial aircraft that meet this restriction, Penguin C from UAV Factory [19] and TARSIS 25 from AERTEC Solutions [20] were selected to set the initial geometric dimensions. However, the wing geometry was simplified, setting a constant chord length airfoil in all the 2 m wingspan with an aspect ratio ($\mathcal{R}$) equal to

10. A 200 mm SD7003 airfoil was chosen as the main airfoil due to its low parasitic drag at low and medium Reynolds numbers (*Re*). This airfoil is widely studied in aerodynamic research, and it is easy to find experimental and computational data in the literature to validate the simulations performed using it, for example, in [21–24].

Table 1 includes a summary of all the relevant aerodynamic parameters for this study. This includes the parasitic drag coefficient due to the fuselage, empennage and the other aircraft elements, not including the wing, $C_{D,0,\text{extra}}$, and also the Oswald efficiency factor, *e*. With these parameters, the total drag *D* can be computed as in Equation (1):

$$D = \frac{1}{2} \cdot \rho_\infty \cdot U_\infty^2 \cdot S \cdot \left( C_{D,0,\text{wing}} + C_{D,0,\text{extra}} + \frac{C_L^2}{\pi \cdot \mathit{AR} \cdot e} \right) \tag{1}$$

where $\rho_\infty$ is the far-field air density, $U_\infty$ is the upstream wind speed, *S* is the wing surface, $C_{D,0,\text{wing}}$ is the parasitic drag coefficient of the wing and $C_L$ is the lift coefficient. The values of $C_{D,0,\text{wing}}$ and $C_{D,0,\text{extra}}$ are computed using geometrical information of aircraft with a similar mission, including the mentioned Penguin C and TARSIS 25, as well as the Harmon and Hiserote's aircraft [4,5]. The Oswald efficiency factor *e* is estimated using the methods described in [25,26].

**Table 1.** Aerodynamic, design parameters and engine data.

| Design parameters | |
|---|---|
| Aspect ratio | 10 |
| Wing area | $0.4\,\text{m}^2$ |
| Wingspan | 2 m |
| Wing chord | 0.2 m |
| Maximum takeoff mass | 25 kg |
| **Aerodynamic parameters** | |
| $C_{D,0,\text{extra}}$ (fuselage, empennage, others) | 0.011 |
| Oswald efficiency factor (*e*) | 0.8 |

Accordingly, the DA4052 double blade propeller designed by UIUC [27] is selected. Complete blade geometry and wind tunnel testing data of this propeller is available, so the simulation results can be validated.

## 3. Methods

In this section, he computational domain is presented besides the design variables involved in the optimisation and selection of the best DEP and BLI case. Then the different computational models used in order to calculate the performance of different RPAS configurations are presented. First, the computational fluid dynamics (CFD) method used to compute the series hybridisation with DEP and BLI is explained. CFD simulations are performed with different configurations: A section of the wing, a single propeller and a section of the wing with a propeller in the trailing edge. These simulations serve as input to a model to compute the range of the aircraft in different conditions.

### 3.1. Computational Setup

All the simulations are composed of a large horseshoe domain, a wing section and an actuator disc that simulates the propeller. The actuator disc will be discussed in Section 3.3. The described domain can be observed in Figure 1, where the main dimensions and mesh are displayed.

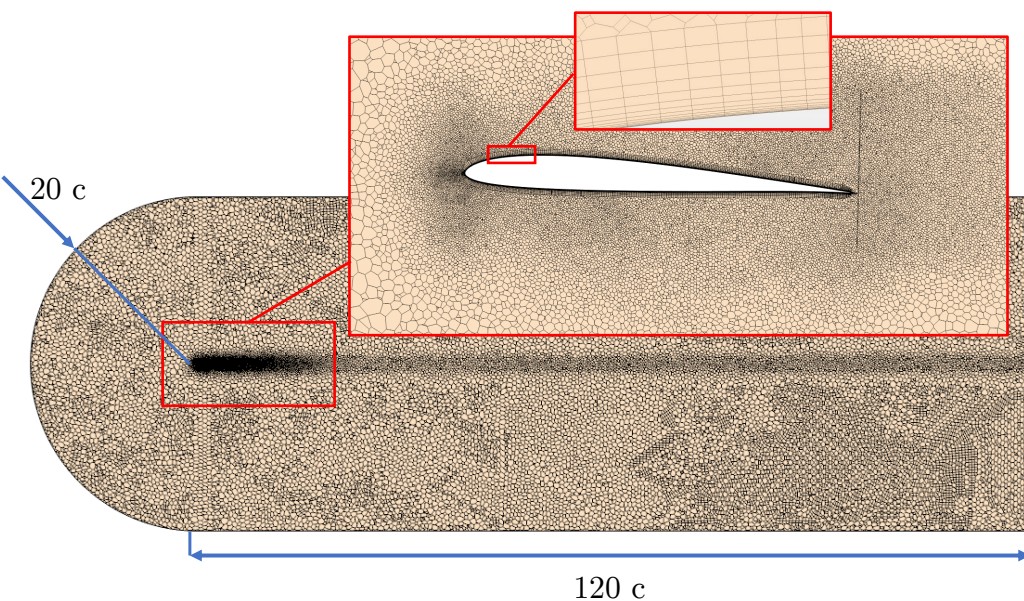

20 c

120 c

**Figure 1.** Sketch of the computational grid used for the current calculations.

Using this domain, the upwind boundary condition is set 20 chords away from the wing with a free-stream speed imposed on it. This is also valid for the turbulence intensity and length scale of the flow. Downstream of the wing, the boundary is set at 120 chords as a static pressure outlet. The boundary in the wall of the wing is set as smooth roughness wall with non-slip conditions. The remaining boundaries are modeled assuming that normal derivatives of the variables are zero at these locations, assuming they are far from the body. Boundaries above and below the wing are separated 80 chords in order to ensure that their location do not disturb the results.

To verify that the dimensions of the domain are enough and do not interfere in the solution, a domain independence analysis was performed. Extra cases were carried out, doubling and halving the vertical distance between exterior boundaries. The variation in lift coefficient, $C_L$, and parasitic drag coefficient, $C_{D,0}$ between the chosen case and the largest domain was less than 1%. Hence, it is considered that the solution obtained is independent of the size of the domain used.

All CFD simulations consist of a three-dimensional wing section with an actuator disc near the airfoil trailing edge. This actuator disc simulates the propeller by means of Blade Element Momentum Theory (BEMT). The described airfoil and propeller are sketched in Figure 2 with their main dimensions.

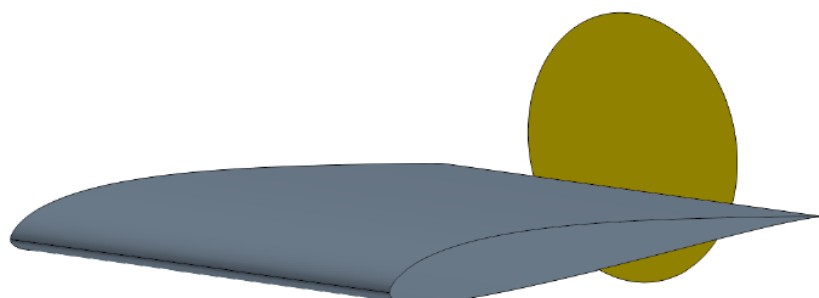

**Figure 2.** Section of wing simulated with virtual disc.

For each radius, different vertical positions are used in the simulation, which are given by the relative distance between the trailing edge an the center of the actuator disc. The highest position corresponds to 100%, where the propeller is all above the trailing edge, being 0% the lowest position where the shaft of the propeller is aligned with the trailing

edge of the airfoil, as can be seen in Figure 3. In this figure, the described positions are shown looking at the wing with a viewing axis parallel to the chord.

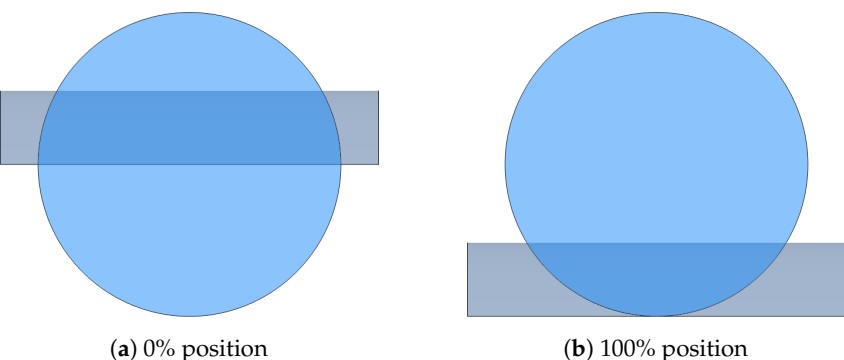

(**a**) 0% position    (**b**) 100% position

**Figure 3.** Maximum and minimum propeller position above the trailing edge.

There is a constant length gap between the trailing edge and the actuator disc equal to 1% of the chord length. This gap should be kept as small as possible because, as the airfoil wake moves downstream, it dissipates due to viscous shear stresses, therefore more power will be needed to move the propeller that ingest the wake. As explained in [28], to increase the propulsive efficiency using BLI, the ingestion must take place before the wake dissipates, what is achieved using an small gap.

Next section presents the methods and models used to compute the performance characteristics of the RPAS with different configurations.

### 3.2. CFD Methodology

All the CFD simulation are numerically solved by mean of a Finite Volume Method using the commercial software Simcenter STAR-CCM+ with steady-state Reynolds-Average Navier-Stokes (RANS) equations approach. Second order methods were used for solving the advection and diffusive terms.

The Spalar–Allmaras turbulence model is chosen to solve the Reynolds stress tensor. This is a one-equation model designed for external aerodynamic applications such as the present case and used extensively in DEP and BLI research [8,29].

As the Mach number is kept below 0.2 in all the simulated cases, the flow can be modelled as incompressible. All domain is meshed with a polyhedral mesh except for the boundary layer around the wing, where a prismatic mesh with a geometric grow distribution is used. The total thickness of the prismatic mesh is 3 mm divided in 14 layers, ensuring an $y^+$ minor to one in 99% of the wall, able to resolve the viscous subrange of the boundary layer.

The complete mesh is sketched at Figure 1, where different refinement zones and a detail near the airfoil boundary layer can be observed.

A mesh independence analysis has been carried out with an arbitrary angle of attack of 4.8° and a Reynolds number of $3 \times 10^5$. For estimating the value of the drag coefficient as function of the base size used in the independence, a generalized Richardson extrapolation was performed, as described by [30]. Final base size was set at 1 mm, guaranteeing an error of less than 2% in drag coefficient. The complete final mesh is composed of more than $3.3 \times 10^6$ elements.

### 3.3. Actuator Disc Setup

As already mentioned in Section 3.1, the propeller is modelled using an actuator disc (also known as virtual disc) approach coupled with a blade-element theory submodel. This approach is much less expensive in simulation time compared to simulating a real propeller and is widely used in DEP and BLI research, as can be seen in [8,14].

Using the blade-element theory (BEMT), the propeller is divided into different sections along the entire blade radius, resulting in two dimensional wings which must be aerody-

namically characterised for the model. A tip loss correction factor is fixed as constant and equal to 0 at a relative span of 0.97 due to the three dimensional aerodynamic behaviour of the last fraction of the length of the blade, as advised in [31]. Simplifying the propeller geometry to an actuator disk and solving it by means of the BEMT has some potential issues that have to be taken into account in order to assess the quality of the results. The accuracy of this approach is reduced as sections of the propeller work in stall or transonic conditions. Moreover, the detailed flow behaviour around sections of the propeller blades can not be simulated with this method, so only the results upstream and downstream of the actuator disk are valid: In the actuator disk itself, all the effects are collapsed into a zero-thickness surface. As the present work does not look at the performance parameters in stall or transonic propeller conditions and the detailed flow around the blade airfoils is not studied, the limitations of the actuator disk approach do not affect the main results and conclusions.

The geometry of the blade used is specified in [32] in detail and the different aerodynamic coefficients needed for each section are calculated using a potential flow method with interactive boundary layer correction under XFLR5 [33], which is based on XFOIL [34].

Using the mesh specified in the previous section, the resolution of the actuator disc is fixed to 8 elements in both azimuth and radial directions using an uniform distribution. To carry out the validation of the use of an actuator disc as model for the propeller, the same propeller is simulated with the same domain without any wing thus avoiding the effects of DEP and BLI. The results are compared against experimental data of [32] and shown in Figure 4.

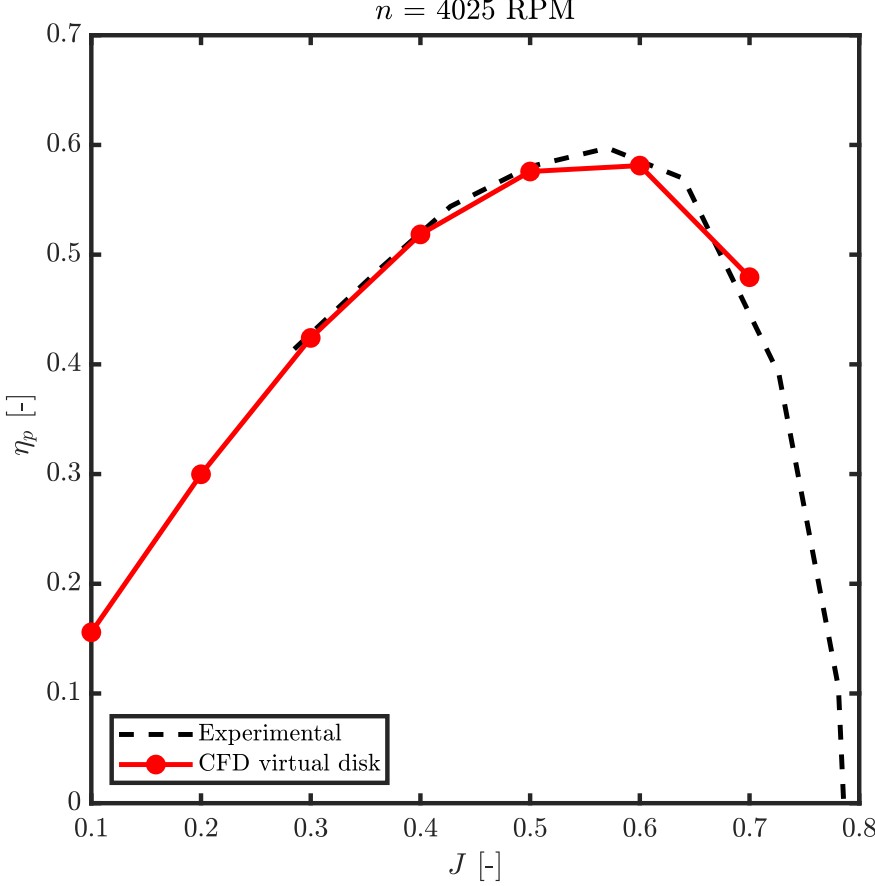

**Figure 4.** CFD and experimental propulsive efficiency comparison at 4025 rpm.

In Figure 4, the propulsive efficiency $eta_p$ for both the simulated and the experimental propeller are compared in a common range of advance ratio $J$, where these terms are defined in Equations (2) and (3).

$$\eta_p = \frac{T \cdot U_\infty}{P} \tag{2}$$

where $T$ is the propeller thrust, $P$ is the propeller power and $U_\infty$ the air velocity.

$$J = \frac{U_\infty}{n \cdot 2 \cdot r_{\text{propeller}}} \tag{3}$$

where $n$ is the propeller rotational speed and $r_{\text{propeller}}$ its radius. When expressing the air speed in units of distance divided by seconds, it is customary to express $n$ in Hz.

Fair agreement was found between BEMT + CFD and the experimental results, as both result in similar propulsive efficiency, so all the DEP and BLI cases will be modeled using this approach.

All DEP and BLI cases consist of a single actuator disc, therefore the rotational speed must be set to take into account the thrust produced by all the propellers distributed along the wing.

For simplicity, individual thrust will be considered the same on each propeller, since the flight is considered stationary and leveled. This way, the rotational speed of the propeller can be set in a way that the total thrust produced by all of them would be equal to the total drag of the aircraft in each simulation.

The total drag produced by the entire aircraft is estimated from the simulation force coefficients, taking into account extra parasitic drag coefficient, $C_{D,0,\text{extra}}$, due the non wing produced drag, as shown in Section 2.

In all the simulations, the rotational speed of the propeller is controlled so the total thrust of the aircraft, $T \cdot n_{\text{propellers}}$, is set equal to the total drag, $D$, as in Equation (4):

$$T \cdot n_{\text{propellers}} = D = \frac{1}{2} \cdot \rho_\infty \cdot U_\infty^2 \cdot S \cdot \left( C_{D,0,\text{wing}} + C_{D,0,\text{extra}} + \frac{C_L^2}{\pi \cdot A\!R \cdot e} \right) \tag{4}$$

where the propeller thrust, $T$, is multiplied by the total number of propellers, $n_{\text{propellers}}$. As only one propeller is simulated, the total number of propellers results in a function of the domain width. Since the wingspan is known and all engines are assumed to be equally spaced, the distribution of propellers is uniquely defined. In all the simulations in this document, a total of 13 propellers is considered. Given that the thrust produced by the propellers is equal to the aircraft drag in all the cases, the simulation data can be used and interpolated to predict the performance of the aircraft when flying with different weights and load factors, assuming straight-and-level flight or level turns.

## 4. Results and Discussion

This section presents the different results of the study. It is divided into two parts: The first one, in which the results of the simulations for different actuator disc positions, at different angles of attack and maintaining the Reynolds number are analysed, so that an optimal design position can be discerned from both the propulsive and aerodynamic point of view; then, the computational fluid dynamics (CFD) analysis of the best case is computed and compared with the base case, showing the improvement of the different efficiencies from an aerodynamic point of view. Each case used 128 cpu-hours to converge.

### 4.1. Propeller Position Cfd Analysis

CFD simulations for different propeller positions and angles of attack are studied, keeping the same blade radius and distribution. This way, all cases have a propeller radius of 40 mm, a distribution of 13 engines on the wing and a draft angle of 1.5°, only changing the position of the actuator disc over the trailing edge as a geometric parameter.

In this way, it is possible to quantify the effect of the propeller position for different angles of attack on the different aerodynamic and propulsive coefficients. Five different propeller heights relative to the trailing edge, as explained in Section 3.3, are studied for a constant Reynolds equal to $3 \times 10^5$. To decide which is the optimal propeller position, the product of propulsive efficiency and aerodynamic efficiency is represented in Figure 5. Bigger distances measured in the direction of the chord were also tested: Increasing them 4% of the chord decreased the aerodynamic and propulsive efficiency product by 4%, whereas incresing this distance a further 4% of the chord decreased the efficiencies product an extra 0.5%.

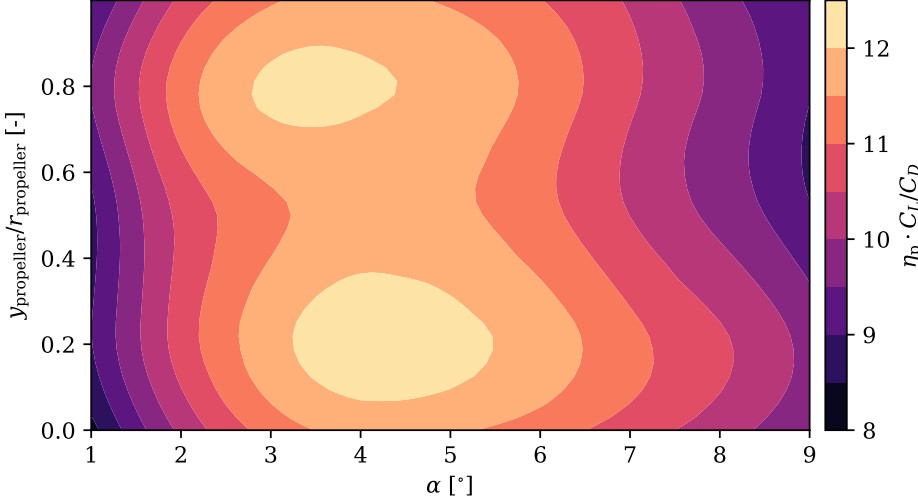

**Figure 5.** Aerodynamic and propulsive efficiency for each propeller position above the trailing edge and angle of attack.

Figure 5 shows two different positions that optimise the product of efficiencies for a wide range of angles of attack. This is due to the fact that two positions maximise this product but optimising the two efficiencies differently. The improvement in this overall performance in the higher position is due to the improvement in aerodynamic efficiency, while the improvement found in the lowest positions is produced due to the improvement in propulsive efficiency.

To understand how efficiencies are maximised, they are represented separately in Figures 6 and 7.

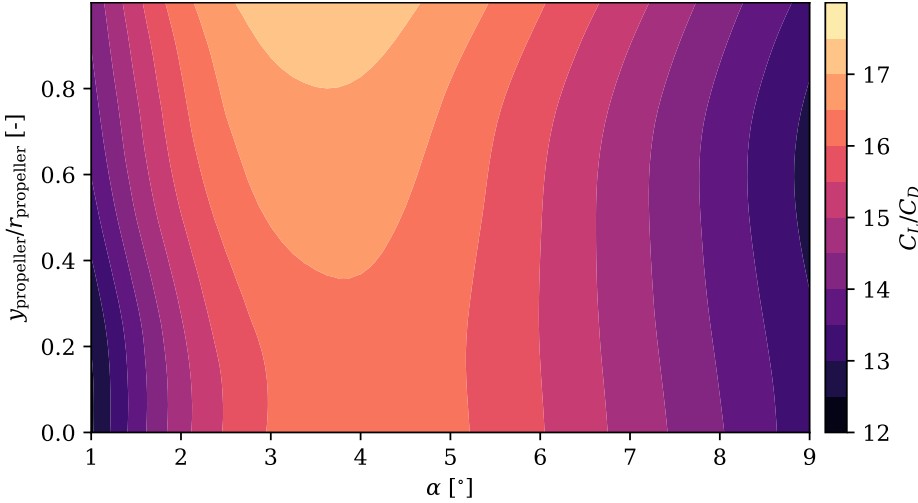

**Figure 6.** Aerodynamic for each propeller position above the trailing edge and angle of attack.

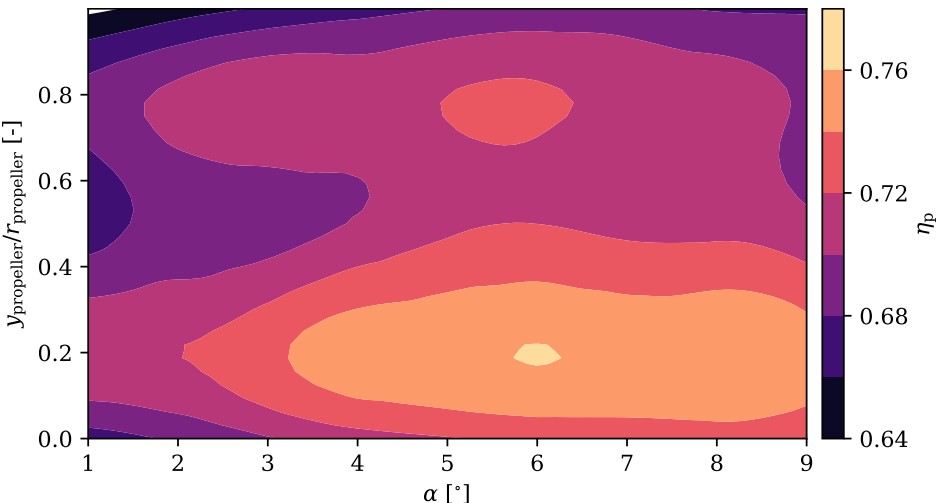

**Figure 7.** Propulsive efficiency for each propeller position above the trailing edge and angle of attack.

Starting with the aerodynamic efficiency, it increases as the propeller position raises in a wide range of angles of attack. The position of the propeller changes the circulation around the wing section. As this position is higher, the lift increases at the same time as the parasitic drag decreases. This is reflected in the pressure coefficient where if this coefficient is represented in Figure 8 for all positions and a fixed angle, it can be seen that the suction peak is greater as the position moves upwards.

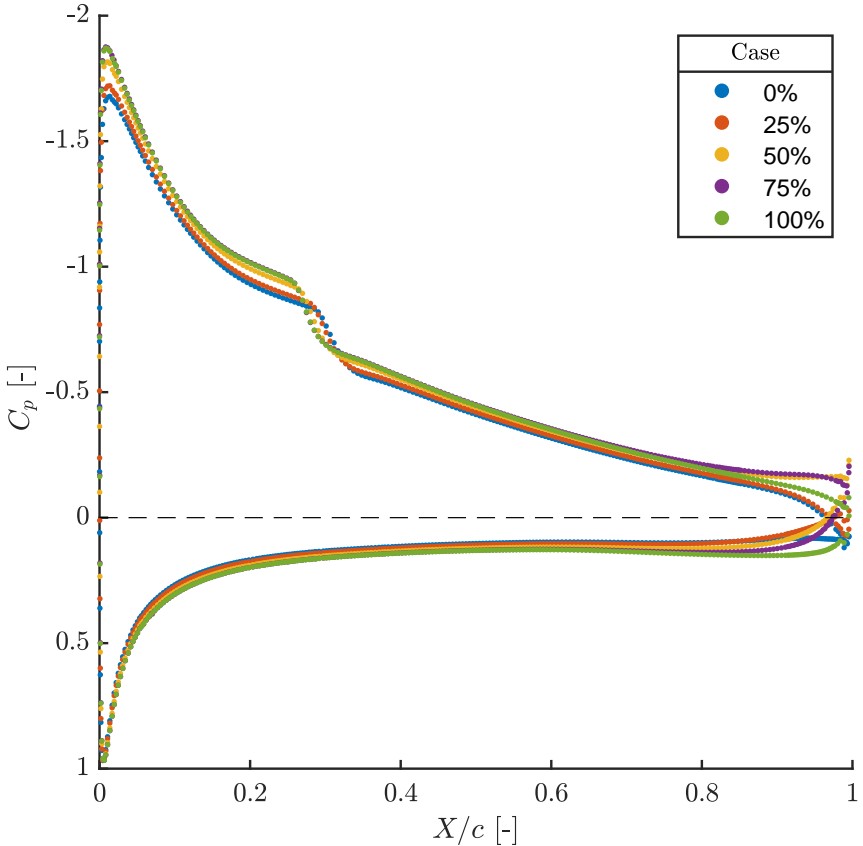

**Figure 8.** Pressure coefficient around the airfoil at 5° of angle of attack for each propeller position relative to the trailing edge.

The effect of raising the propeller over the trailing edge translates in the pressure coefficient $C_p$ into an effect similar to increasing in the apparent angle of attack, as when

flying inside the upwash of other wing. In addition, the elevation of the propeller produces less reacceleration of the pressure side, since a smaller portion of the propeller remains below the trailing edge.

The friction coefficient is plotted in Figure 9. It can be seen that, as the propeller position is higher, the friction in the last part of the suction side increases, thus increasing the parasitic drag associated with this parameter. It should be noted that, in the highest position, the effect of the propeller decreases, which is reflected in a drop in the friction coefficient $C_f$.

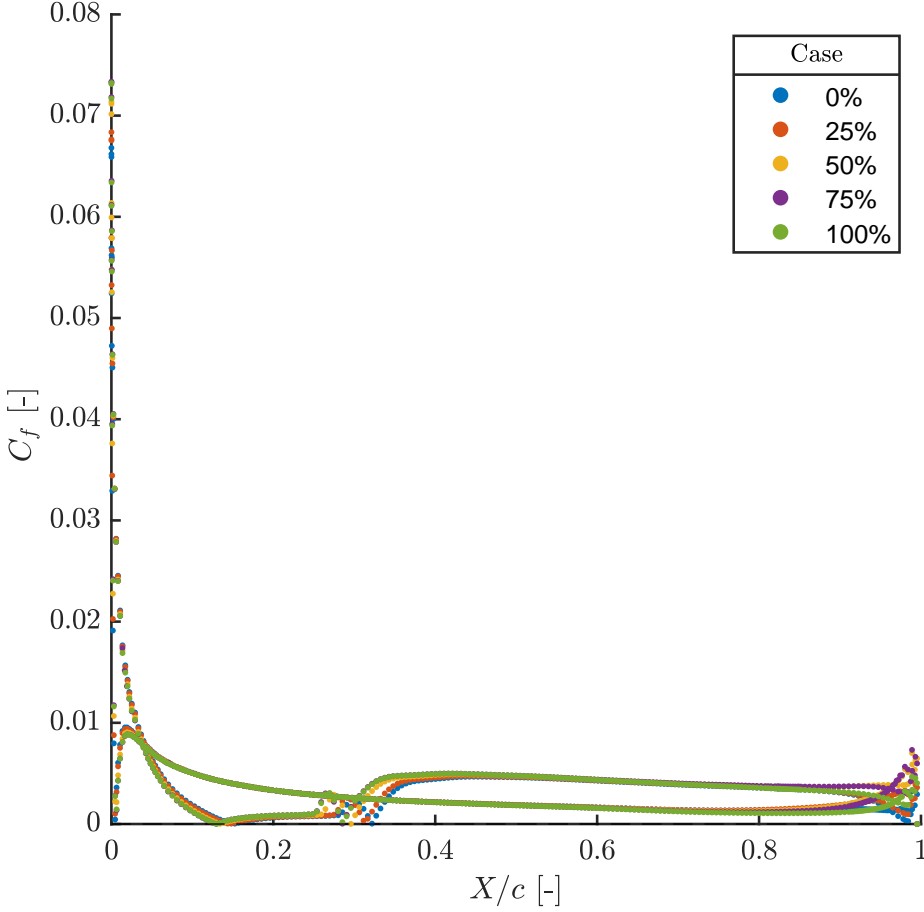

**Figure 9.** Friction coefficient around the airfoil at 5° of angle of attack for each propeller position relative to the trailing edge.

In the pressure side, a result similar to that of $C_p$ can be obtained, where a higher propeller position implies a lower re-acceleration of the flow and, therefore, a lower $C_f$.

The effect of the propeller position on the propulsive efficiency results in the combination of several phenomena. The expectation would be to see an increase in this efficiency as the propeller is in the lower positions, that is, more centered on the trailing edge. The propulsive efficiency increase due to boundary layer ingestion can be summed up in that the propeller ingests a flow region whose speed is less than the flight speed, decreasing the power needed to be produced by the propeller for the same required force. As the propeller is positioned lower above the trailing edge, the boundary layer portion that will affect the propeller will be higher. This way, the propulsive efficiency increases. However, since the simulations are not carried out at the same lift coefficient but at the same air speed, the operating point at the same angle of attack varies for the different positions. The change in aerodynamic efficiency translates into different thrust requests for each position, which varies the rotational speed of the propellers. By varying the rotational speed while maintaining the air speed, the advance parameter $J$ changes so that in each

position the propeller will work at a different operating point. While the effect of the operation is less than the purely geometric aspect, in this way a high propulsive efficiency is still obtained also with higher propeller positions, but being always greater in the lower positions, with an optimum around the 20 % position.

### 4.2. Best Case Cfd Analysis

In this study a case with a propeller radius of 40 mm, a distribution of 13 engines on the wing and a draft angle of 1.5° with respect to the airfoil chord is presented. The position above the trailing edge is fixed at 31.5 %. This case maximizes the product of aerodynamic efficiency and propulsive efficiency.

CFD results in near-cruise conditions (i.e., $35 \, \mathrm{m \, s^{-1}}$) are analysed, setting for the study an angle of attack of 3° and a Reynolds number equal to $5 \times 10^5$. As the angle of attack is low, the boundary layer is fully attached to the airfoil and its growth is moderate as can be seen in Figure 10a, representing the velocity contours in a mid-plane around the wing.

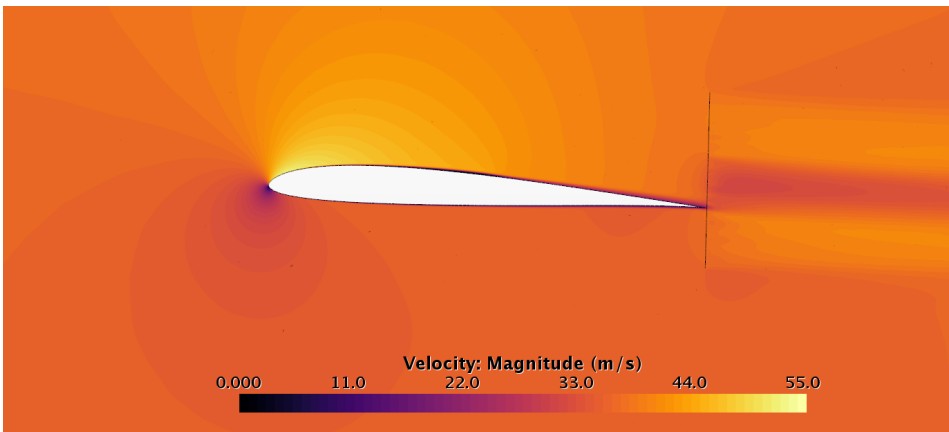

(**a**) Velocity contours in DEP BLI optimal case

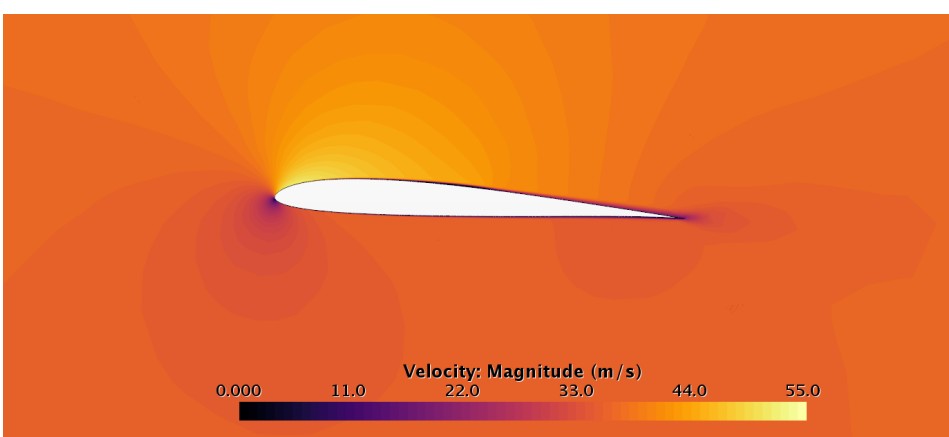

(**b**) Velocity contours in base case

**Figure 10.** *U* contours in mid-plane for best DEP BLI configuration and base case comparison at an angle of attack of 3° and a Reynolds number of $5 \times 10^5$.

If the BLI case in Figure 10a and the base case in Figure 10b are compared, speed similarities can be observed. Due to the propeller, the case with BLI has a greater reacceleration in the pressure side, at the same time that speeds up the wake behind the propeller. In addition, speed isolines have been added in Figure 11 showing a zoom of the pressure side for a better comparison.

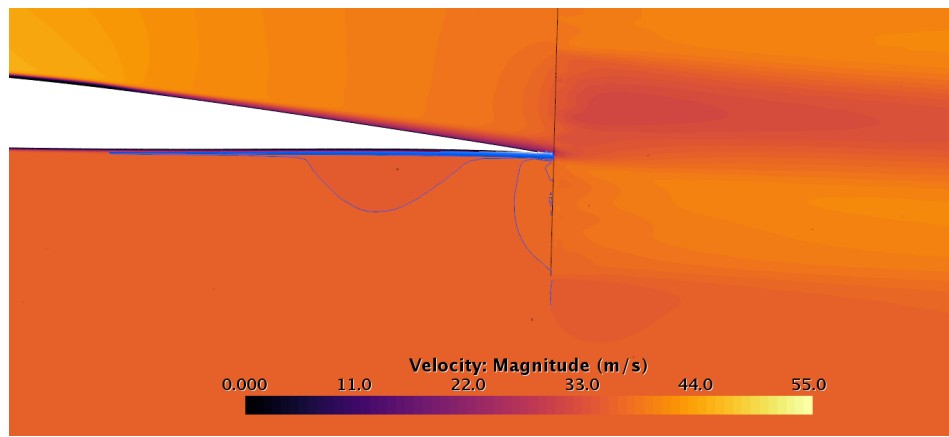

(**a**) Detail of velocity contours in DEP BLI optimal case

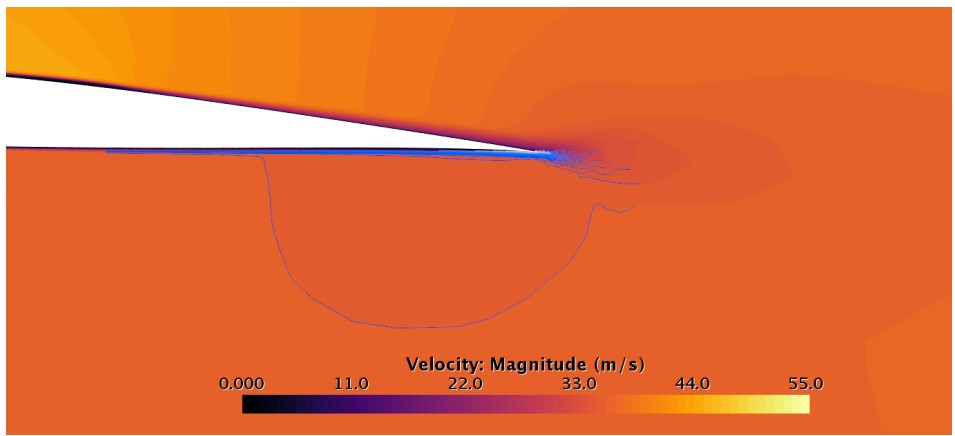

(**b**) Detail of velocity contours in base case

**Figure 11.** Detail of $U$ in midplane for best DEP BLI configuration and base case with isovelocity lines under the pressure side for better comparison.

In order to compare the product between aerodynamic and propulsive efficiency in both cases, an additional simulation is carried out including only the propeller. This analysis is carried out in this way since in the base case the propulsive system is considered decoupled from the aerodynamic performance of the wing. In the simulation, the same flow velocity and advance parameter $J$ from the DEP-BLI case is imposed.

The aerodynamic and propulsive data for both cases are collected in Table 2. The case with BLI has a higher $C_L$ than the base case, however this increase is accompanied by an increase in $C_{D,0}$ that impairs aerodynamic efficiency by 1%. Nevertheless, the boundary layer bathed area of the actuator disc increases the propulsive efficiency of the BLI case by 8%, improving the product of efficiencies.

**Table 2.** Coefficient comparison between optimal DEP and BLI case and base case at an angle of attack of 3° and a Reynolds number of $5 \times 10^5$.

|  | $C_L$ | $C_{D,0,\text{wing}}$ | $C_L/C_D$ | $\eta_p$ | $C_L/C_D \cdot \eta_p$ |
|---|---|---|---|---|---|
| Base case | 0.484 | 0.00769 | 17.280 | 0.692 | 11.963 |
| DEP BLI case | 0.505 | 0.00841 | 17.080 | 0.748 | 12.773 |

To understand the differences in the aerodynamic coefficients, the pressure coefficient $C_p$ of the case without BLI is plotted against the simulation proposed in Figure 12 in different wing spans.

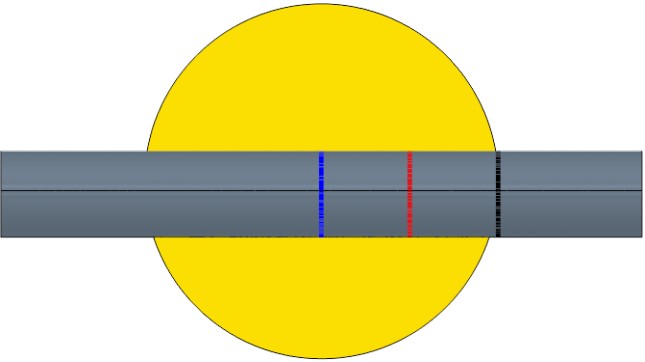

(**a**) Span positions in frontal view

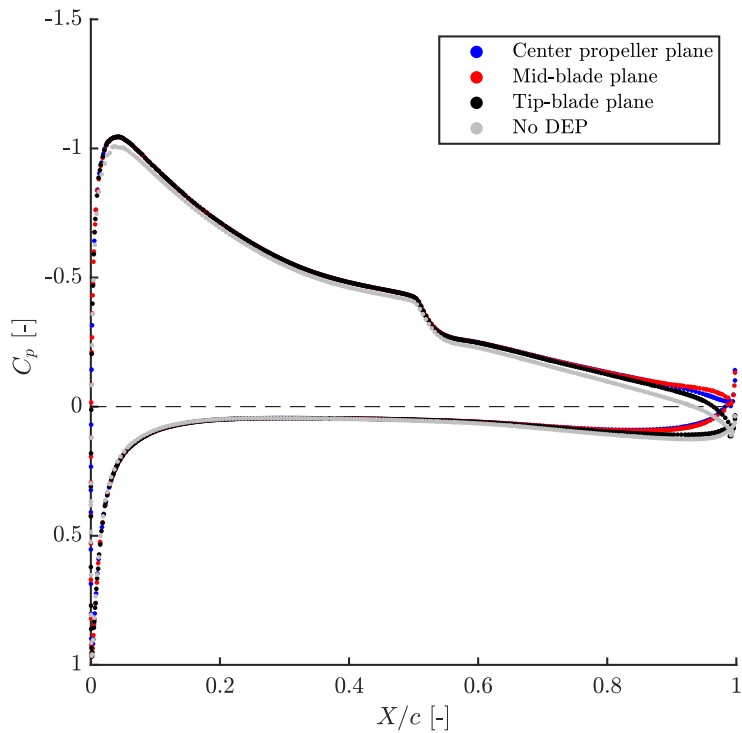

(**b**) Pressure coefficient distribution at different spans

**Figure 12.** *Cp* comparison for best DEP BLI configuration in different wing spans and base case middle plane.

Figure 12 shows that the suction peak magnitude in the BLI case is 4% higher than the base case, with increased suction on the entire domain. Simultaneously, in the first half of the pressure side, the case with BLI has a bigger $C_p$ value than the wing alone, which translates to an increase in the lift coefficient.

At 50% of the chord, a transitional laminar separation bubble (LSB) can be observed in the BLI configuration, upstream compared with the base case. The effect of placing the propeller on the trailing edge is similar to increasing the angle of attack of the airfoil since the LSB moves upstream as the angle of attack increases, a common trend that can be observed experimentally, as shown in [27]. The effect is, thus, similar to that of flying with upwash: The resultant of the pressure forces is tilted towards the leading edge, lift is increased and the boundary layer transition appears closer to the leading edge.

In order to confirm the appearance of LSB, the friction coefficient $C_f$ is plotted in Figure 13 for both cases in the mid span.

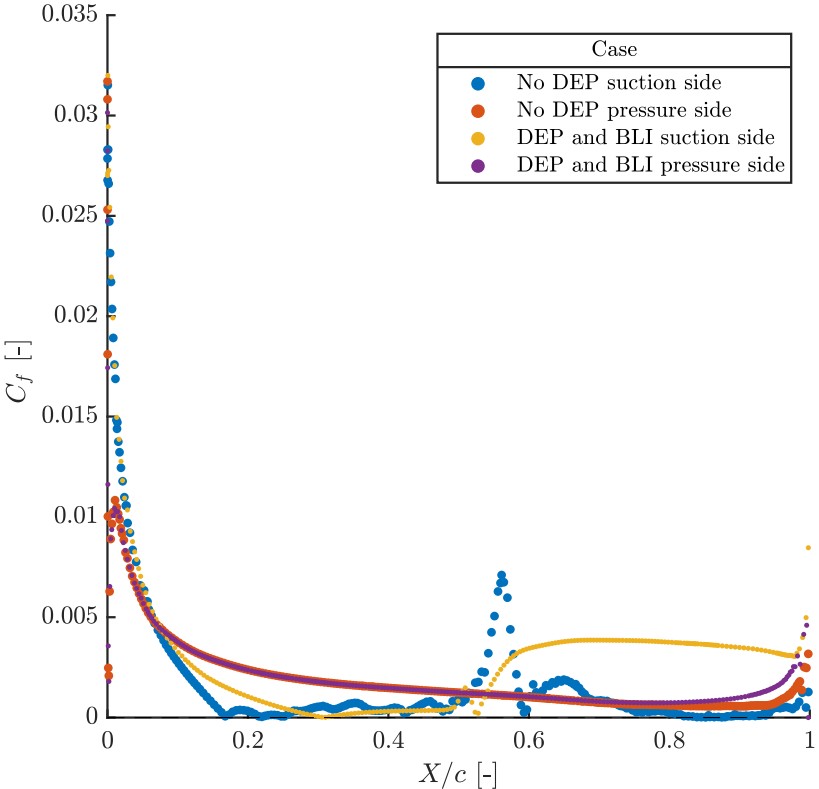

**Figure 13.** $C_f$ comparison for DEP BLI case and base case without DEP and BLI distinguishing between suction side and pressure side.

Comparing the $C_f$ in the suction side of both cases, it can be verified that the LSB appears earlier in the DEP BLI case resulting in a smaller bubble. As Sutton explains in [24], a smaller laminar separation bubble changes the effective inviscid shape of the flow around the wing, decreasing the pressure drag and allowing higher suction peaks to be achieved. However, as Tepperin has found in [15], the decrease in pressure drag can be outweighed by the increase in friction drag. The air suction near the trailing edge creates a lower pressure zone before the propeller accelerates the flow. Flow acceleration will increase the shear force and, in the same way, the friction drag increases, what can be observed after the appearance of LSB in the case of DEP BLI. Identically, since a part of the propeller disc re-accelerates the flow in the pressure side due to its position, $C_f$ is also higher here.

Near the trailing edge, a discrepancy in $C_p$ is appreciated due to how the surface is influenced by the actuator disc in the BLI case. To understand this effect, the pressure coefficient in Figure 12 is represented in three spans that cross the surface of the wing in the direction of flow: The blue one passes through the propeller separating it in half, the red through half its radius and the black plane corresponding to the blade edge.

The discrepancy of pressure coefficients between the spans can only be seen from 60% of the chord: Upstream, the presence of the propeller has the same effect on the entire surface of the airfoil.

In the outer spans away from the mid-plane, the decrease in pressure is less important and begins to approach the base case. As the effects of the propeller in the behaviour of the flow near the trailing edge are maximised close to its center, its influence in the boundary layer detachment at high angles of attack is expected to present non-uniformities across the wing span.

## 5. Conclusions

An RPAS with distributed electric propulsion and boundary layer ingestion configuration is feasible employing an hybrid electric power plant, a fuel cell or batteries, so it is a

technological configuration that is applicable in an almost powerplant-agnostic manner. This novel configuration leads to improvements in both aerodynamic and propulsive efficiency compared to a classical distribution with one propeller without boundary layer ingestion. These efficiency improvements are achieved as long as the different geometrical parameters of the propeller distribution are set appropriately.

In this work, a methodology to analyse a small, 25 kg aircraft with a DEP BLI configuration has been discussed, whereby using a few simulations of a portion of the wing with a single rotor, the best distribution and overall performance of the aircraft has been estimated. The numerical study has been performed by a three-dimensional finite-volume simulation of the domain around a wing section coupled with a Blade Element Model Theory actuator disc to include the propeller. The rotational speed of the propellers has been set so their thrust equals the total drag of the aircraft, and the simulations have been performed for different angles of attack and vertical separations between the propeller shaft and the wing trailing edge. The simulations have been also carried out for the propeller and the wing separately, validating them against data found in the literature and comparing them with the DEP BLI cases. Results such as the pressure coefficient, skin friction coefficient, aerodynamic efficiency and propulsive efficiency have been obtained for all the simulations, so the overall performance parameters and the main effects of the DEP BLI configuration have been studied. The study of these parameters and effects is the main contribution of this manuscript, as the literature discussing them is still limited for small aircraft.

A distribution of 13 propellers along the wing, with 40 mm blade radius, a draft angle equal to 1.5° and a position of the center of the propeller relative to the trailing edge set at 31.5% has been found to maximise the product of both efficiencies at near-cruise conditions, achieving an improvement of 7%.

The vertical position of the propeller affect in different ways two important performance parameters: The propulsive efficiency and the aerodynamic efficiency. It has been shown that, when the propeller shaft is closer to the trailing edge, the propulsive efficiency is maximised, but there appears some penalties in the aerodynamic efficiency. When the shaft is moved upwards, the propeller section effectively ingesting the boundary layer of the wing is reduced, producing lower values of propulsive efficiency. In that case, however, the form drag of the wing is reduced and an increase in the lift coefficient appears, attaining higher values of aerodynamic efficiency. These opposing trends lead to two different optima: one at a vertical position around 20% of the propeller radius and another one at around 80%.

According to the simulations, moving the propeller upwards reduces its effect in the pressure side of the wing, moves the laminar separation bubble towards the leading edge and increases the suction peak. While the friction coefficient is also increased, the net effect is an increase of the circulation around the wing section and a decrease of the form drag, similar to what may be expected due to upwash effects.

The close proximity of the propellers to the trailing edge of the wing will probably generate interesting challenges due to vibrations, both in the propeller and in the airframe, which is an interesting aspect to study and discuss in future works. Regarding the position of the propellers, it also affects the pitching moment of the whole wing plus propellers group. The final geometry of an aircraft designed with this configuration should be adapted accordingly in order to not affect in a negative way its stability properties.

**Author Contributions:** Conceptualization, L.M.G.-C.; Data curation, P.V.; Formal analysis, P.V.; Funding acquisition, J.R.S.; Investigation, P.V.; Methodology, L.M.G.-C.; Project administration, L.M.G.-C.; Resources, J.R.S.; Software, L.M.G.-C. and P.V.; Supervision, L.M.G.-C.; Validation, L.M.G.-C. and P.V.; Visualization, L.M.G.-C. and P.V.; Writing—original draft, P.V.; Writing—review and editing, J.R.S., A.O.T., L.M.G.-C. and P.V. All authors have read and agreed to the published version of the manuscript.

**Funding:** This research was funded by the Agencia Estatal de Investigación of Spain through grant number PID2020-119468RA-I00/AEI/10.13039/501100011033.

**Institutional Review Board Statement:** Not applicable.

**Informed Consent Statement:** Not applicable.

**Data Availability Statement:** The data presented in this study are available on request from the corresponding author.

**Acknowledgments:** The authors wish to thank P. Raga and P. Quintero for their help with the setup of the computational simulations.

**Conflicts of Interest:** The authors declare no conflict of interest. The funders had no role in the design of the study; in the collection, analyses, or interpretation of data; in the writing of the manuscript, or in the decision to publish the results.

## Abbreviations

The following abbreviations are used in this manuscript:

| Abbreviations | |
|---|---|
| BLI | Boundary layer ingestion |
| BEMT | Blade Element Model Theory |
| *BSFC* | Brake-specific fuel consumption |
| CFD | Computational fluid dynamics |
| DEP | Distributed electrical propulsion |
| EASA | European Union Aviation Safety Agency |
| ERA | Environmentally Responsible Aviation |
| HE | Hybrid electric |
| ICE | Internal combustion engine |
| ITDS | Information Technology Development Solutions |
| LSB | Laminar separation bubble |
| NASA | National Aeronautics and Space Administration |
| RANS | Reynols-averaged Navier-Stokes |
| RPAS | Remotely piloted aircraft system |
| SESAR | Single European Sky ATM Research |
| TE | Trailing edge |
| UAV | Unmanned aerial vehicle |
| Roman letters | |
| $\mathcal{R}$ | Aspect ratio |
| $b$ | Wingspan |
| $c$ | Chord |
| $C_L$ | Lift coefficient |
| $C_D$ | Drag coefficient |
| $C_{D,0,\text{extra}}$ | Parasitic drag coefficient of the aircraft without the wing |
| $C_{D,0,\text{wing}}$ | Parasitic drag coefficient of the wing |
| $C_p$ | Pressure coefficient |
| $C_f$ | Friction coefficient |
| $D$ | Drag |
| $e$ | Oswald efficiency factor |
| $J$ | Advance ratio |
| $n$ | Rotational speed |
| $P$ | Power |
| $r_{\text{propeller}}$ | Propeller radius |
| $Re$ | Reynolds |
| $S$ | Wing surface |
| $T$ | Thrust |
| $U_\infty$ | Air speed |
| $X$ | Position across the chord |

| $y_{propeller}$ | Position of the propeller shaft above the trailing edge |
| Greek letters | |
| $\alpha$ | Angle of attack |
| $\eta_p$ | Propulsion efficiency |
| $\rho$ | Density |

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
