# Peer review of "Computational Study of the Propeller Position Effects in Wing-Mounted, Distributed Electric Propulsion with Boundary Layer Ingestion in a 25 kg Remotely Piloted Aircraft"

_drones, doi:10.3390/drones5030056_

Round 1

Reviewer 1 Report

This manuscript studies the effect of changing the propellers position in the aerodynamic performance parameters of distributed electric propulsion with boundary layer ingestion system in a 25 kg fixed-wing aircraft, as well as in the performance of the propellers. Different computational models have been carried out in order to calculate the performance of different remotely piloted aircraft system configurations. First, the computational fluid dynamics (CFD) method has been used in order to compute the series hybridization with distributed electrical propulsion (DEP) and boundary layer ingestion (BLI) is explained. CFD simulations have been carried out with different configurations: a section of the wing, a single propeller and a section of the wing with a propeller in the trailing edge. Grid sensitivity calculation has been carried out. Final base size was set at 1mm, 178 guaranteeing an error of less than 2%. Fair agreement was found between BEMT + CFD and the experimental results. The computational results show the trade-offs between the aerodynamic efficiency and the propeller efficiency when the vertical position is varied. According to the simulations, moving the propeller upwards reduces its effect in the pressure side of the wing, moves the laminar separation bubble towards the leading edge and increases the suction peak. The friction coefficient is also increased. The authors should perform minor revisions on this interesting paper before it can be considered for publication in Drones journal.

Comments and Suggestions for Authors

1) (Introduction section - Lines no. 37 - 38). Not all the readers are familiar with Boundary Layer Ingestion (BLI). It is recommended to cite the following paper:

Budziszewski, N.; Friedrichs, J. Modelling of A Boundary Layer Ingesting Propulsor, Energies 2018, 11, 708. https://doi.org/10.3390/en11040708.

2) (Introduction section – Lines no. 21). Some of the abbreviations used in the manuscript are not shown in the abbreviation list. For example the abbreviation EASA should be included in this list.

3) (Line no. 178) what is the meaning of “said coefficient”?

4) (Line no. 220 Equation (4)) what is the total number of propellers used in this computational work?

5) (Line no. 270 Table 2) the authors should mention the value of the angle of attack (α) in this table.

6) Figure 9 shows the propulsive efficiency as a function of the propeller position. It shows that the propulsive efficiency increases with decreasing the distance between the propellers. Does it mean that the number of propellers is increased?

7) (Figure 12 – graph of the pressure coefficient) in my opinion the negative values of the pressure coefficients (located in the y axis) should be located under zero and not above it.

8) (Reference section) the year should be bold (See the following reference example):

Author 1, A.B.; Author 2, C.D. Title of the article. Abbreviated Journal Name Year, Volume, page range, DOI.

9) (Reference no. 32 and 33) the addresses of the websites should be provided.

Author Response

As the corrections made in the manuscript following the comments of the 3 reviewers may be of interest to Reviewer 1,  the authors have included the whole rebuttal in one single document. Please see attachment.

Reviewer 2 Report

It is an interesting article whcih focuses on a trendy topic in the UAV sector. I would suggest the following to improve its quality.

1) Assess the gap between propeller and wing TE, in order to see the performance affection due to this allocation variable

2) I would prefer the whole computational setup in the methods section to easier the reading of the manuscript

3) It is necessary to describe the effect of syimplifying the propeller geometry to the actuator disk and DEP to Actuator disk, although it is this addressed to reference 8,13 I think for this specific case it is necessary the authors explain further the effects, as this is a cornerstone in the CFD results they present after.

Line 128, it looks informal (don't)

Figure 13 legend is in spanish

References regarding BLI and distributed propusion are quite old, there are new articles that have made more updated predictions. I recommend to update some of them

Author Response

The rebuttal includes also the responses to the other 4 reviewers, as they may also be interesting for Reviewer 2.

Reviewer 3 Report

The paper presents an analysis In this work, of a small 25 kg aircraft with a DEP BLI with a single rotor, to estimate the best distribution and overall performance of the aircraft. A distribution of propellers along the wing has been found to maximize the product at near-cruise conditions, achieving an improvement of 7%. The contribution is clear and brings an important contribution to the field. The paper is well written with clear results and a good presentation.

Some questions:

1. Could the authors present the aircraft design?

2. The modeling of the complete aircraft, at least of its body might change the results? I am wondering about the influence of wind shocking against the aircraft body and returning to the window. 

3. You mention the work targets small UAVs. Is this a limitation? How the approach you presented could be extended and generalized to other aircraft?

4. You might consider, at least in the related works, the discussion about the vibration impact of the propeller in the UAV. Maybe changing its position could affect the stability of the aircraft.

Author Response

As the corrections made in the manuscript following the comments of the 3 reviewers may be of interest to Reviewer 2  the authors have included the whole rebuttal in one single document. Please see attachment.

Reviewer 4 Report

The paper presents an analysis of a small 25 kg aircraft with DEP. The optimal location of the center of the propeller relative to the trailing edge of the wing and for its vertical position was found, and the improvements of the optimal locations were shown. The paper is well written and organized. I have a few minor comments:

  1. Can the presented methodology be scaled up? If yes, what are the possibilities for it and what are the limitations?
  2. What is the case if the aircraft is not assumed to be rigid? How can the methodology be extended to flexible aircraft?
  3. What are the simulation times?
  4. Can the results be supported by higher fidelity simulations as well?

I recommend revising the paper based on these comments.

Author Response

The rebuttal includes also the responses to the other 4 reviewers, as they may also be interesting for Reviewer 4.

Reviewer 5 Report

The paper is well written. The contributions are clearly defined in the introduction, therefore not leaving the reader not knowing what it is. Typical errors and problems that would usually have been identified thus far with reading through it, has not been seen. There are a few small issues that just needs to be clarified and corrected...

  • Near-cruise speeds - what speeds is cruise speed? This value varies between aircraft.
  • Results and discussion sections should be joined if possible. Initially it was thought that there were results missing that would be analyzed in the discussion, only to find these results in the discussion section. So this missing information is then introduced in the discussion, but the graphs are also as part of the results.
  • Within the conclusion section, the contribution can be re-iterated, as to how it was achieved in the paper (maybe indicate this more specifically), to allow for a reader that might be searching for papers to identify the summary of the paper with the contributions, will be able to pick this up very quickly when reading the conclusion.
  • Finally, this study is interesting, but how does the CFD analysis compare with that of actual system. In other words, if you placed the prop on the wing, and implemented the required power and speed, do you get the a lift force that might be similar tot hat of the simulations? Just a way to integrate the computer and real-life situation....

Author Response

As the corrections made in the manuscript following the comments of the 3 reviewers may be of interest to Reviewer 3,  the authors have included the whole rebuttal in one single document. Please see attachment.

Round 2

Reviewer 2 Report

Line 237 is missing a citation

The other comments have been addressed satisfactorily by the authors.